# Reinforcement Learning with Automated Auxiliary Loss Search

**Tairan He**[1*] **Yuge Zhang**[2] **Kan Ren**[2†] **Minghuan Liu**[1]
**Che Wang**[3] **Weinan Zhang**[1] **Yuqing Yang**[2] **Dongsheng Li**[2]
[1]Shanghai Jiao Tong University    [2]Microsoft Research Asia    [3]New York University
whynot@sjtu.edu.cn    kan.ren@microsoft.com

## Abstract

A good state representation is crucial to solving complicated reinforcement learning (RL) challenges. Many recent works focus on designing auxiliary losses for learning informative representations. Unfortunately, these handcrafted objectives rely heavily on expert knowledge and may be sub-optimal. In this paper, we propose a principled and universal method for learning better representations with auxiliary loss functions, named Automated Auxiliary Loss Search (A2LS), which automatically searches for top-performing auxiliary loss functions for RL. Specifically, based on the collected trajectory data, we define a general auxiliary loss space of size $7.5 \times 10^{20}$ and explore the space with an efficient evolutionary search strategy. Empirical results show that the discovered auxiliary loss (namely, `A2-winner`) significantly improves the performance on both high-dimensional (image) and low-dimensional (vector) unseen tasks with much higher efficiency, showing promising generalization ability to different settings and even different benchmark domains. We conduct a statistical analysis to reveal the relations between patterns of auxiliary losses and RL performance. The codes and supplementary materials are available at https://seqml.github.io/a2ls.

## 1 Introduction

Reinforcement learning (RL) has achieved remarkable progress in games [31, 47, 50], financial trading [8] and robotics [13]. However, in its core part, without designs tailored to specific tasks, general RL paradigms are still learning implicit representations from critic loss (value predictions) and actor loss (maximizing cumulative reward). In many real-world scenarios where observations are complicated (e.g., images) or incomplete (e.g., partial observable), training an agent that is able to extract informative signals from those inputs becomes incredibly sample-inefficient.

Therefore, many recent works have been devoted to obtaining a good state representation, which is believed to be one of the key solutions to improve the efficacy of RL [23, 24]. One of the main streams is adding auxiliary losses to update the state encoder. Under the hood, it resorts to informative and dense learning signals in order to encode various prior knowledge and regularization [40], and obtain better latent representations. Over the years, a series of works have attempted to figure out the form of the most helpful auxiliary loss for RL. Quite a few advances have been made, including observation reconstruction [51], reward prediction [20], environment dynamics prediction [40, 6, 35], etc. But we note two problems in this evolving process: (i) each of the loss designs listed above are obtained through empirical trial-and-errors based on expert designs, thus heavily relying on human labor and expertise; (ii) few works have used the final performance of RL as an optimization objective to directly search the auxiliary loss, indicating that these designs could be sub-optimal.

---

*The work was conducted during Tairan He's internship at Microsoft Research.
†The corresponding author is Kan Ren.

36th Conference on Neural Information Processing Systems (NeurIPS 2022).

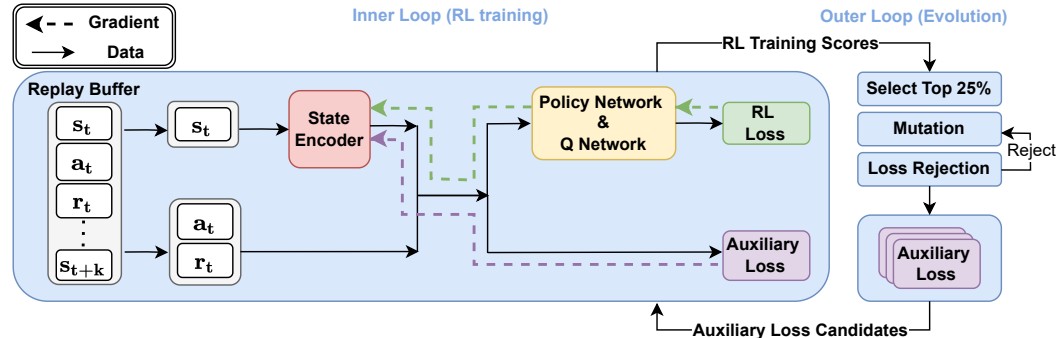

Figure 1: Overview of A2LS. A2LS contains an inner loop (left) and an outer loop (right). The inner loop performs an RL training procedure with searched auxiliary loss functions. The outer loop searches auxiliary loss functions using an evolutionary algorithm to select the better auxiliary losses.

To resolve the issues of the existing handcrafted solution mentioned above, we decide to automate the process of designing the auxiliary loss functions of RL and propose a principled solution named Automated Auxiliary Loss Search (A2LS). A2LS formulates the problem as a bi-level optimization where we try to find the best auxiliary loss, which, to the most extent, helps train a good RL agent. The outer loop searches for auxiliary losses based on RL performance to ensure the searched losses align with the RL objective, while the inner loop performs RL training with the searched auxiliary loss function. Specifically, A2LS utilizes an evolutionary strategy to search the configuration of auxiliary losses over a novel search space of size $7.5 \times 10^{20}$ that covers many existing solutions. By searching on a small set of simulated *training environments* of continuous control from Deepmind Control suite (DMC) [43], A2LS finalizes a loss, namely `A2-winner`.

To evaluate the generalizability of the discovered auxiliary loss `A2-winner`, we test `A2-winner` on a wide set of *test environments*, including both image-based and vector-based (with proprioceptive features like positions, velocities and accelerations as inputs) tasks. Extensive experiments show the searched loss function is highly effective and largely outperforms strong baseline methods. More importantly, the searched auxiliary loss generalizes well to unseen settings such as (i) different robots of control; (ii) different data types of observation; (iii) partially observable settings; (iv) different network architectures; and (v) even to a totally different discrete control domain (Atari 2600 games [1]). In the end, we make detailed statistical analyses on the relation between RL performance and patterns of auxiliary losses based on the data of whole evolutionary search process, providing useful insights on future studies of auxiliary loss designs and representation learning in RL.

## 2   Problem Formulation and Background

We consider the standard Markov Decision Process (MDP) $\mathcal{E}$ where the state, action and reward at time step $t$ are denoted as $(s_t, a_t, r_t)$. The sequence of rollout data sampled by the agent in the episodic environment is $(s_0, \ldots, s_t, a_t, r_t, s_{t+1}, \cdots, s_T)$, where $T$ represents the episode length. Suppose the RL agent is parameterized by $\omega$ (either the policy $\pi$ or the state-action value function $Q$), with a state encoder $g_\theta$ parameterized by $\theta \subseteq \omega$ which plays a key role for representation learning in RL. The agent is required to maximize its cumulative rewards in environment $\mathcal{E}$ by optimizing $\omega$, noted as $\mathcal{R}(\omega; \mathcal{E}) = \mathbb{E}_\pi[\sum_{t=0}^{T-1} r_t]$.

In this paper, we aim to find the optimal auxiliary loss function $\mathcal{L}_{\text{Aux}}$ such that the agent can reach the best performance by optimizing $\omega$ under a combination of an arbitrary RL loss function $\mathcal{L}_{\text{RL}}$ together with an auxiliary loss $\mathcal{L}_{\text{Aux}}$. Formally, our optimization goal is:

$$\max_{\mathcal{L}_{\text{Aux}}} \quad \mathcal{R}(\min_{\omega} \mathcal{L}_{\text{RL}}(\omega; \mathcal{E}) + \lambda \mathcal{L}_{\text{Aux}}(\theta; \mathcal{E}); \mathcal{E}) , \tag{1}$$

where $\lambda$ is a hyper-parameter balancing the relative weight of the auxiliary loss. The left part (inner loop) of Figure 1 illustrates how data and gradients flow in RL training when an auxiliary loss is enabled. Some instances of $\mathcal{L}_{\text{RL}}$ and $\mathcal{L}_{\text{Aux}}$ are given in Appendix B. Unfortunately, existing auxiliary losses $\mathcal{L}_{\text{Aux}}$ are handcrafted, which heavily rely on expert knowledge, and may not generalize well

Table 1: Typical solution with auxiliary loss and their common elements.

| Auxiliary Loss | Operator | Input Elements | | |
|---|---|---|---|---|
| | | Horizon | Source | Target |
| Forward dynamics [35, 40, 6] | MSE | 1 | $\{s_t, a_t\}$ | $\{s_{t+1}\}$ |
| Inverse dynamics | MSE | 1 | $\{a_t, s_{t+1}\}$ | $\{s_t\}$ |
| Reward prediction [20, 6] | MSE | 1 | $\{s_t, a_t\}$ | $\{r_t\}$ |
| Action inference [40, 6] | MSE | 1 | $\{s_t, s_{t+1}\}$ | $\{a_t\}$ |
| CURL [23] | Bilinear | 1 | $\{s_t\}$ | $\{s_t\}$ |
| ATC [42] | Bilinear | k | $\{s_t\}$ | $\{s_{t+1}, \cdots, s_{t+k}\}$ |
| SPR [39] | N-MSE | k | $\{s_t, a_t, a_{t+1}, \cdots, a_{t+k-1}\}$ | $\{s_{t+1}, \cdots, s_{t+k}\}$ |

in different scenarios as shown in the experiment part. To find better auxiliary loss functions for representation learning in RL, we introduce our principled solution in the following section.

## 3 Automated Auxiliary Loss Search

To meet our goal of finding top-performing auxiliary loss functions without expert assignment, we turn to the help of automated loss search, which has shown promising results in the automated machine learning (AutoML) community [27, 28, 48]. Correspondingly, we propose Automated Auxiliary Loss Search (A2LS), a principled solution for resolving the above bi-level optimization problem in Equation 1. A2LS resolves the inner problem as a standard RL training procedure; for the outer one, A2LS defines a finite and discrete search space (Section 3.1), and designs a novel evolution strategy to efficiently explore the space (Section 3.2).

### 3.1 Search Space Design

We have argued that almost all existing auxiliary losses require expert knowledge, and we expect to search for a better one automatically. To this end, it is clear that we should design a search space that satisfies the following desiderata.

- **Generalization**: the search space should cover most of the existing handcrafted auxiliary losses to ensure the searched results can be no worse than handcrafted losses;

- **Atomicity**: the search space should be composed of several independent dimensions to fit into any general search algorithm [30] and support an efficient search scheme;

- **Sufficiency**: the search space should be large enough to contain the top-performing solutions.

Given the criteria, we conclude and list some existing auxiliary losses in Table 1 and find their commonalities, as well as differences. We realize that these losses share similar components and computation flow. As shown in Figure 2, when training the RL agent, the loss firstly selects a sequence $\{s_t, a_t, r_t\}_{t=i}^{i+k}$ from the replay buffer, when $k$ is called *horizon*. The agent then tries to predict some elements in the sequence (called *target*) based on another picked set of elements from the sequence (called *source*). Finally, the loss calculates and minimizes the prediction error (rigorously defined with *operator*). To be more specific, the encoder part $g_\theta$ of the agent, first encodes the *source* into latent representations, which is further fed into a predictor $h$ to get a prediction $y$; the auxiliary loss is computed by the prediction $y$ and the target $\hat{y}$ that is translated from the *target* by a target encoder $g_{\hat{\theta}}$, using an *operator* $f$. The target encoder is updated in an momentum manner as shown in Figure 2 (details are given in Appendix C.1.2). Formally,

$$\mathcal{L}_{\text{Aux}}(\theta; \mathcal{E}) = f\Big( h\big(g_\theta(\text{seq}_{\text{source}})\big), g_{\hat{\theta}}(\text{seq}_{\text{target}})\Big), \tag{2}$$

where $\text{seq}_{\text{source}}, \text{seq}_{\text{target}} \subseteq \{s_t, a_t, r_t\}_{t=i}^{i+k}$ are both subsets of the candidate sequence. And for simplicity, we will denote $g_\theta(s_t, a_t, r_t, s_{t+1}, \cdots)$ as short for $[g_\theta(s_t), a_t, r_t, g_\theta(s_{t+1}), \cdots]$ for the rest of this paper (the encoder $g$ only deals with states $\{s_i\}$). Thereafter, we observe that these existing auxiliary losses differ in two dimensions, i.e., *input elements* and *operator*, where *input elements* are further combined by *horizon*, *source* and *target*. These differences compose our search dimensions of the whole space. We then illustrate the search ranges of these dimensions in detail.

**Input elements.** The *input elements* denote all inputs to the loss functions, which can be further

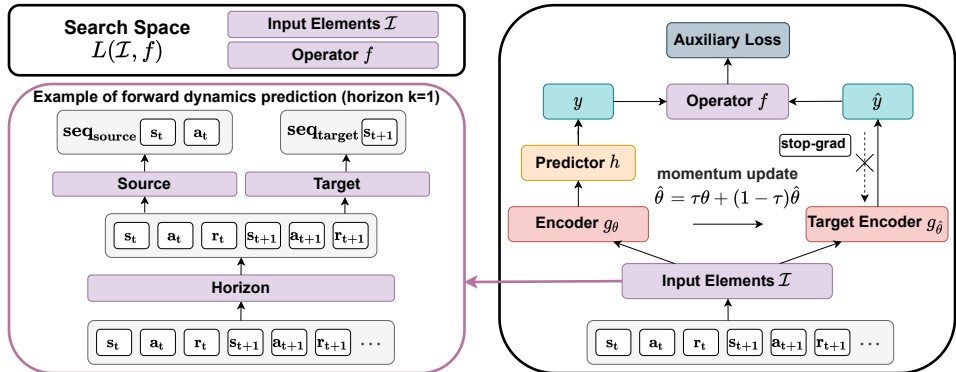

Figure 2: Overview of the search space $\{\mathcal{I}, f\}$ and the computation graph of auxiliary loss functions. $\mathcal{I}$ selects a candidate sequence $\{s_t, a_t, r_t\}_{t=i}^{i+k}$ with *horizon $k$*; then determine a *source* and a *target* as arbitrary subsets of the sequence; an encoder $g_\theta$ first encodes the *source* into latent representations, which is fed into a predictor $h$ to get a prediction $y$; the auxiliary loss is computed over the prediction $y$ and the ground truth $\hat{y}$ that is translated from the *target* by a target encoder $g_{\hat{\theta}}$, using a operator $f$.

disassembled as *horizon*, *source* and *target*. Different from previous automated loss search works, the *target* here is not "ground-truth" because auxiliary losses in RL have no labels beforehand. Instead, both *source* and *target* are generated via interacting with the environment in a self-supervised manner. Particularly, the *input elements* first determine a candidate sequence $\{s_t, a_t, r_t\}_{t=i}^{i+k}$ with *horizon $k$*. Then, it chooses two subsets from the candidate sequence as *source* and *target* respectively. For example, the subsets can be $\{s_t\}, \{s_t, s_{t+1}\}$, or $\{s_t, r_{t+1}, a_{t+2}\}, \{s_t, s_{t+1}, a_{t+1}\}$, etc.

**Operator.** Given a prediction $y$ and its target $\hat{y}$, the auxiliary loss is computed by an operator $f$, which is often a similarity measure. In our work, we cover all different operators $f$ used by the previous works, including inner product (Inner) [17, 42], bilinear inner product (Bilinear) [23], cosine similarity (Cosine) [3], mean squared error (MSE) [35, 6] and normalized mean squared error (N-MSE) [39]. Additionally, other works also utilize contrastive objectives, e.g., InfoNCE loss [33], incorporating the trick to sample un-paired predictions and targets as negative samples and maximize the distances between them. This technique is orthogonal to the five similarity measures mentioned above, so we make it optional and create $5 \times 2 = 10$ different operators in total.

**Final design.** In the light of preceding discussion, with the definition of *input elements* and *operator*, we finish the design of the search space, which satisfactorily meets the desiderata mentioned above. Specifically, the space is **generalizable** to cover most of the existing handcrafted auxiliary losses; additionally, the **atomicity** is embodied by the compositionality that all *input elements* work with any *operator*; most importantly, the search space is **sufficiently** large with a total size of $7.5 \times 10^{20}$ (detailed calculation can be found in Appendix E) to find better solutions.

## 3.2   Search Strategy

The success of evolution strategies in exploring large, multi-dimensional search space has been proven in many works [19, 4]. Similarly, A2LS adopts an evolutionary algorithm [37] to search for top-performing auxiliary loss functions over the designed search space. In its essence, the evolutionary algorithm (i) keeps a population of loss function candidates; (ii) evaluates their performance; (iii) eliminates the worst and evolves into a new better population. Note that step (ii) of "evaluating" is very costly because it needs to train the RL agents with dozens of different auxiliary loss functions. Therefore, our key technical contribution contains how to further reduce the search cost (Section 3.2.1) and how to make an efficient search procedure (Section 3.2.2).

### 3.2.1   Search Space Pruning

In our preliminary experiment, we find out the dimension of *operator* in the search space can be simplified. In particular, MSE outperforms all the others by significant gaps in most cases. So we effectively prune other choices of *operators* except MSE. See Appendix D.1 for complete comparative results and an ablation study on the effectiveness of search space pruning.

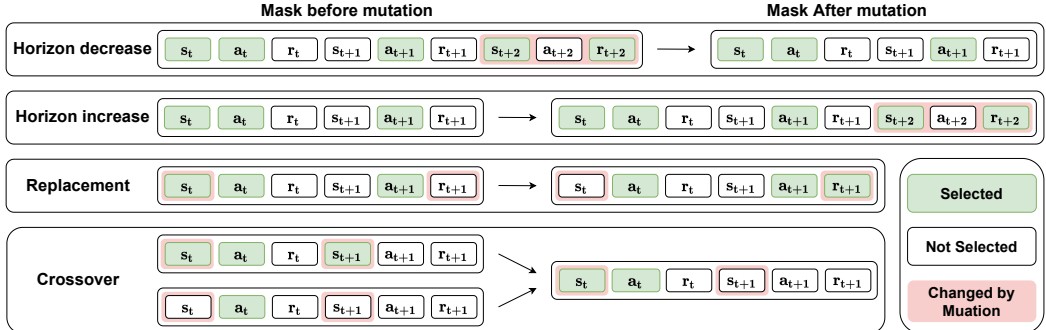

Figure 3: Four types of mutation strategy for evolution. We represent both the *source* and the *target* of the input elements as a pair of binary masks, where each bit of the binary mask represents *selected* (green block) by 1 or *not selected* (white block) by 0.

### 3.2.2 Evolution Procedure

Our evolution procedure roughly contains four important components: (i) **evaluation and selection**: a population of candidate auxiliary losses is evaluated through an inner loop of RL training, then we select the top candidates for the next evolution stage (i.e., generation); (ii) **mutation**: the selected candidates mutate to form a new population and move to the next stage; (iii) **loss rejection**: filter out and skip evaluating invalid auxiliary losses for the next stage; and (iv) **bootstrapping initial population**: assign more chance to initial auxiliary losses that may contain useful patterns by prior knowledge for higher efficiency. The step-by-step evolution algorithm is provided in Algorithm 1 in the appendix, and an overview of the A2LS pipeline is illustrated in Figure 1. We next describe them in detail.

**Evaluation and selection.** At each evolution stage, we first train a population of candidates with a population size $P = 100$ by the inner loop of RL training. The candidates are then sorted by computing the approximated *area under learning curve* (AULC) [11, 41], which is a single metric reflecting both the convergence speed and the final performance [46] with low variance of results. After each training stage, the top-25% candidates are selected to generate the population for the next stage. We include an ablation study on the effectiveness of AULC in Appendix D.3.

**Mutation.** To obtain a new population of auxiliary loss functions, we propose a novel mutation strategy. First, we represent both the *source* and the *target* of the input elements as a pair of binary masks, where each bit of the mask represents *selected* by 1 or *not selected* by 0. For instance, given a candidate sequence $\{s_t, a_t, r_t, s_{t+1}, a_{t+1}, r_{t+1}\}$, the binary mask of this subset sequence $\{s_t, a_t, r_{t+1}\}$ is denoted as 110001. Afterward, we adopt four types of mutations, also shown in Figure 3: (i) replacement (50% of the population): flip the given binary mask with probability $p = \frac{1}{2 \cdot (3k+3)}$ with the horizon length $k$; (ii) crossover (20%): generate a new candidate by randomly combining the mask bits of two candidates with the same horizon length in the population; (iii) horizon decrease and horizon increase (10%): append new binary masks to the tail or delete existing binary masks at the back. (iv) random generation (20%): every bit of the binary mask is generated from a Bernoulli distribution $\mathcal{B}(0.5)$.

**Loss rejection protocol.** Since the auxiliary loss needs to be differentiable with respect to the parameters of the state encoder, we perform a gradient flow check on randomly generated loss functions during evolution and skip evaluating invalid auxiliary losses. Concretely, the following conditions must be satisfied to make a valid loss function: (i) having at least one state element in seq$_{\text{source}}$ to make sure the gradient of auxiliary loss can propagate back to the state encoder; (ii) seq$_{\text{target}}$ is not empty; (iii) the horizon should be within a reasonable range ($1 \leq k \leq 10$ in our experiments). If a loss is rejected, we repeat the mutation to fill the population.

**Bootstrapping initial population.** To improve the computational efficiency so that the algorithm can find reasonable loss functions quickly, we incorporate prior knowledge into the initialization of the search. Particularly, before the first stage of evolution, we bootstrap the initial population with a prior distribution that assigns high probability to auxiliary loss functions containing useful patterns like dynamics and reward prediction. More implementation details are provided in Appendix C.3.

# 4 Evolution and Searched Results

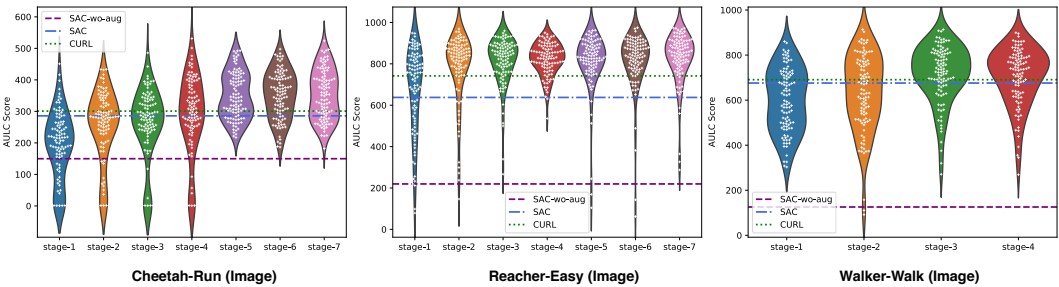

Figure 4: Evolution process in the three training (image-based) environments. Every white dot represents a candidate of auxiliary loss, and y-axis shows its corresponding approximated AULC score [11, 41]. The horizontal lines show the scores of the baselines. The AULC score is approximated with the average evaluation score at 100k, 200k, 300k, 400k, 500k time steps.

As mentioned in Section 1, we expect to find auxiliary losses that align with the RL objective and generalize well to unseen *test environments*. To do so, we use A2LS to search over a small set of *training environments*, and then test the searched results on a wide range of *test environments*. In this section, we first introduce the evolution on *training environments* and search results.

## 4.1 Evolution on Training Environments

The *training environments* are chosen as three image-based (observations for agents are images) continuous control tasks in DMC benchmark [43], Cheetah-Run, Reacher-Easy, and Walker-Walk. For each environment, we set the total budget to 16k GPU hours (on NVIDIA P100) and terminate the search when the resource is exhausted. Due to computation complexity, we only run one seed for each inner loop RL training, but we try to prevent such randomness by cross validation (see Section 4.2). We use the same network architecture and hyperparameters config as CURL [23] (see Appendix C.4.1 for details) to train the RL agents. To evaluate the population during evolution, we measure A2LS as compared to SAC, SAC-wo-aug, and CURL, where we randomly crop images from $100 \times 100$ to $84 \times 84$ as data augmentation (the same technique used in CURL[23]) for all methods except SAC-wo-aug. The whole evolution process on three environments is demonstrated in Figure 4. Even in the early stages (e.g., stage 1), some of the auxiliary loss candidates already surpass baselines, indicating the high potential of automated loss search. The overall AULC scores of the population continue to improve when more stages come in (detailed numbers are summerized in Appendix D.10). Judging from the trend, we believe the performances could improve even more if we had further increased the budget.

## 4.2 Searched Results: `A2-winner`

Although some candidates in the population have achieved remarkably AULC scores in the evolution (Figure 4), they were only evaluated with one random seed in one environment, making their robustness under question. To ensure that we find a consistently-useful auxiliary loss, we conduct a cross validation. We first choose the top 5 candidates of stage-5 of the evolution on Cheetah-Run (detailed top candidates during the whole evolution procedure are provided in Appendix F). For each of the five candidates, we repeat the RL training on all three *training environments*, shown in Figure 5. Finally, we mark the best among five (green bar in Figure 5) as our final searched result. We call it `A2-winner`, which has the following form:

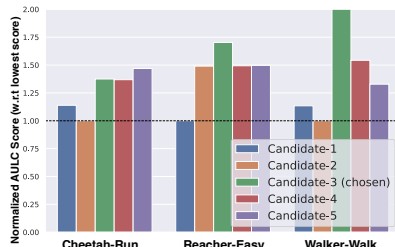

Figure 5: Cross validation on image-based *training environments*.

$$\mathcal{L}_{\text{Aux}}(\theta; \mathcal{E}) = \left\| h\big(g_\theta(s_{t+1}, a_{t+1}, a_{t+2}, a_{t+3})\big) - g_{\hat{\theta}}(r_t, r_{t+1}, s_{t+2}, s_{t+3}) \right\|_2 . \tag{3}$$

Table 2: Episodic rewards (mean & standard deviation for 10 seeds) on DMC100K (100K time steps) and DMC500K (500K time steps). Note that the optimal score of DMC is 1000 for all environments. The baseline methods are PlaNet [16], Dreamer [15], SAC+AE [51], SLAC [26], image-based SAC [14]. Performance values of all baselines are referred to [23], except for Image SAC. Learning curves of all 12 DMC environments are included in Appendix D.2.

| **500K** Steps Scores | A2-winner | CURL[§] | PlaNet[§] | Dreamer[§] | SAC+AE[§] | SLACv1[§] | Image SAC |
|---|---|---|---|---|---|---|---|
| Cheetah-Run[†] | $613 \pm 39$ | $518 \pm 28$ | $305 \pm 131$ | $570 \pm 253$ | $550 \pm 34$ | **$640 \pm 19$** | $99 \pm 28$ |
| Reacher-Easy[†] | **$938 \pm 46$** | $929 \pm 44$ | $210 \pm 390$ | $793 \pm 164$ | $627 \pm 58$ | - | $312 \pm 132$ |
| Walker-Walk[†] | **$917 \pm 18$** | $902 \pm 43$ | $351 \pm 58$ | $897 \pm 49$ | $847 \pm 48$ | $842 \pm 51$ | $76 \pm 44$ |
| Finger-Spin* | **$983 \pm 4$** | $926 \pm 45$ | $561 \pm 284$ | $796 \pm 183$ | $884 \pm 128$ | $673 \pm 92$ | $282 \pm 102$ |
| Cartpole-Swingup* | **$864 \pm 19$** | $841 \pm 45$ | $475 \pm 71$ | $762 \pm 27$ | $735 \pm 63$ | - | $344 \pm 104$ |
| Ball in cup-Catch* | **$970 \pm 8$** | $959 \pm 27$ | $460 \pm 380$ | $897 \pm 87$ | $794 \pm 58$ | $852 \pm 71$ | $200 \pm 114$ |
| **100K** Steps Scores | | | | | | | |
| Cheetah-Run[†] | **$449 \pm 34$** | $299 \pm 48$ | $138 \pm 88$ | $235 \pm 137$ | $267 \pm 24$ | $319 \pm 56$ | $128 \pm 12$ |
| Reacher-Easy[†] | **$778 \pm 164$** | $538 \pm 223$ | $20 \pm 50$ | $314 \pm 155$ | $274 \pm 14$ | - | $277 \pm 69$ |
| Walker-Walk[†] | **$510 \pm 151$** | $403 \pm 24$ | $224 \pm 48$ | $277 \pm 12$ | $394 \pm 22$ | $361 \pm 73$ | $127 \pm 28$ |
| Finger-Spin* | **$872 \pm 27$** | $767 \pm 56$ | $136 \pm 216$ | $341 \pm 70$ | $740 \pm 64$ | $693 \pm 141$ | $160 \pm 138$ |
| Cartpole-Swingup* | **$815 \pm 66$** | $582 \pm 146$ | $297 \pm 39$ | $326 \pm 27$ | $311 \pm 11$ | - | $243 \pm 19$ |
| Ball in cup-Catch* | **$862 \pm 167$** | $769 \pm 43$ | $0 \pm 0$ | $246 \pm 174$ | $391 \pm 82$ | $512 \pm 110$ | $100 \pm 90$ |

†: *Training environments*. ∗: Unseen *test environments*. §: Results reported in [23].

# 5 Generalization Experiments

To verify the effectiveness of the searched results, we conduct various generalization experiments on a wide range of *test environments* in depth. Implementation details and more ablation studies are given in Appendix C and Appendix D.

**Generalize to unseen image-based tasks.** We first investigate the generalizability of A2-winner to unseen image-based tasks by training agents with A2-winner on common DMC tasks and compare with model-based and model-free baselines that use different auxiliary loss functions (see Appendix C.5 for details about baseline methods). The results are summarized in Table 2 where A2-winner greatly outperforms other baseline methods on most tasks, including unseen *test environments*. This implies that A2-winner is a robust and effective auxiliary loss for image-based continuous control tasks to improve both the efficiency and final performance.

**Generalize to totally different benchmark domains.** To further verify the generalizability of A2-winner on totally different benchmark domains other than DMC tasks, we conduct experiments on the Atari 2600 Games [1], where we take

Table 3: Mean and Median scores (normalized by human score and random score) achieved by A2LS and baselines on 26 Atari games benchmarked at 100k time-steps (Atari100k).

| Metric | A2-winner | CURL | Eff. Rainbow | DrQ [22] | Random | Human |
|---|---|---|---|---|---|---|
| Mean Human-Norm'd | **0.568** | 0.381 | 0.285 | 0.357 | 0.000 | 1.000 |
| Median Human-Norm'd | **0.317** | 0.175 | 0.161 | 0.268 | 0.000 | 1.000 |

Efficient Rainbow [44] as the base RL algorithm and add A2-winner to obtain a better state representation. Results are shown in Table 3 where A2-winner outperforms all baselines, showing strong evidence of the generalization and potential usages of A2-winner. Note that the base RL algorithm used in Atari is a value-based method, indicating that A2-winner generalizes well to both value-based and policy-based RL algorithms.

**Generalize to different observation types.** To see whether A2-winner (searched in image-based environments) is able to generalize to the environments with different observation types, we test A2-winner on vector-based (inputs for RL agents are proprioceptive features such as positions, velocities and accelerations) tasks of DMC and list the results in Table 4. Concretely, we compare A2-winner with SAC-Identity, SAC and CURL, where SAC-Identity does not have state encoder while the others share the same state encoder architecture (See Appendix C.1.1 and Appendix D.6 for detailed implementations). To our delight, A2-winner still outperforms all baselines in 12 out of 18 environments, showing A2-winner can also benefit RL performance in vector-based observations. Moreover, the performance gain is particularly significant in more complex environments like Humanoid, where SAC barely learns anything at 1000K time steps. In order to get a deeper understanding of this phenomenon, we additionally visualize the Q loss landscape for both methods in Appendix D.7.

Table 4: Episodic rewards (mean & standard deviation for 10 seeds) on DMC100K (easy tasks) and DMC1000K (difficult tasks) with vector inputs.

| 100K Steps Scores | A2-winner | A2-winner-v | SAC-Identity | SAC | CURL |
|---|---|---|---|---|---|
| Cheetah-Run† | **529 ± 76** | 472 ± 30 | 237 ± 27 | 172 ± 29 | 190 ± 32 |
| Finger-Spin* | 790 ± 128 | **837 ± 52** | 805 ± 32 | 785 ± 106 | 712 ± 83 |
| Finger-Turn hard* | 272 ± 149 | 218 ± 117 | **347 ± 150** | 174 ± 94 | 43 ± 42 |
| Cartpole-Swingup* | 866 ± 24 | **877 ± 5** | 873 ± 10 | 866 ± 7 | 854 ± 17 |
| Cartpole-Swingup sparse* | 634 ± 226 | **695 ± 147** | 455 ± 359 | 627 ± 307 | 446 ± 196 |
| Reacher-Easy* | 818 ± 211 | **934 ± 38** | 697 ± 192 | 874 ± 87 | 749 ± 183 |
| Walker-Stand* | 935 ± 32 | **948 ± 7** | 940 ± 10 | 862 ± 196 | 767 ± 104 |
| Walker-Walk* | **932 ± 39** | 906 ± 78 | 873 ± 89 | 925 ± 22 | 852 ± 64 |
| Walker-Run* | **616 ± 52** | 564 ± 45 | 559 ± 34 | 403 ± 43 | 289 ± 61 |
| Ball in cup-Catch* | 964 ± 7 | **965 ± 7** | 954 ± 12 | 962 ± 13 | 941 ± 32 |
| Fish-Upright* | **586 ± 128** | 498 ± 88 | 471 ± 62 | 400 ± 62 | 295 ± 117 |
| Hopper-Stand* | 177 ± 257 | **311 ± 177** | 14 ± 16 | 26 ± 40 | 6 ± 3 |
| **1,000K Steps Scores** | A2-winner-v | A2-winner | SAC-Identity | SAC | CURL |
| Quadruped-Run† | **863 ± 50** | 838 ± 58 | 345 ± 157 | 707 ± 148 | 497 ± 128 |
| Hopper-Hop† | 213 ± 31 | **278 ± 106** | 121 ± 51 | 134 ± 93 | 60 ± 22 |
| Pendulum-Swingup* | 200 ± 322 | **579 ± 410** | 506 ± 374 | 379 ± 391 | 363 ± 366 |
| Humanoid-Stand* | **329 ± 35** | 286 ± 15 | 9 ± 2 | 7 ± 1 | 7 ± 1 |
| Humanoid-Walk* | **311 ± 36** | 299 ± 55 | 16 ± 28 | 2 ± 0 | 2 ± 0 |
| Humanoid-Run* | 75 ± 37 | **88 ± 2** | 1 ± 0 | 1 ± 0 | 1 ± 0 |

†: *Training environments*. ∗: Unseen *test environments*.

**Generalize to different hypothesis spaces.** The architecture of a neural network defines a hypothesis space of functions to be optimized. During the evolutionary search in Section 4.1, the encoder architecture has been kept static as a 4-layer convolutional neural network. Since encoder architecture may have a large impact on the RL training process [34, 2], we test `A2-winner` with three encoders with different depth of neural networks. The result is shown in Figure 6. Note that even though the auxiliary loss is

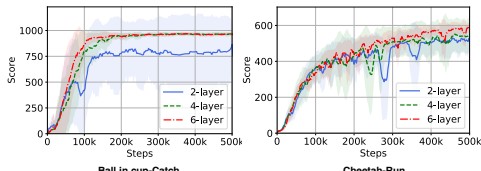

Figure 6: Comparison of `A2-winner` with different depth of convolutional encoder in image-based DMC environments.

searched with a 4-layer encoder, the 6-layer convolutional encoder is able to perform better in both two environments. This proves that the auxiliary loss function of `A2-winner` is able to improve RL performance with a deeper and more expressive image encoder. Moreover, the ranking of RL performance (6-layer > 4-layer > 2-layer) is consistent across the two environments. This shows that the auxiliary loss function of `A2-winner` does not overfit one specific architecture of the encoder.

**Generalize to partially observable scenarios.** Claiming the generality of a method based on conclusions drawn just on fully observable environments like DMC is very dangerous. Therefore, we conduct an ablation study on the Partially Observable Markov Decision Process (POMDP) setting to see whether `A2-winner` is able to perform well in POMDP. We random mask 20% of the state dimensions (e.g., 15 dimensions -> 12 dimensions) to form a POMDP environment in DMC. As demonstrated in Figure 7,

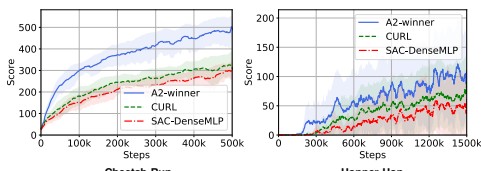

Figure 7: Comparison of `A2-winner` and baselines in partially observable vector-based DMC environments.

`A2-winner` consistently outperforms CURL and SAC-DenseMLP in the POMDP setting in Hopper-Hop and Cheetah-Run, showing that `A2-winner` is not only effective in fully observable environments but also partially observable environments.

**To search or not?** As shown above, the searched result `A2-winner` can generalize well to all kinds of different settings. A natural question here is, however, for a new type of domain, why not perform a new evolution search, instead of simply using the previously searched result? To compare these two solutions, we conduct another evolutionary search similar to Section 4.1 but replaced the three image-based tasks with three vector-based ones (marked by † in Table 4) from scratch. More details are summarized in Appendix D.5. We name the searched result as "`A2-winner-v`". As shown in Table 4, `A2-winner-v` is a very strong-performing loss for vector-based tasks, even stronger than `A2-winner`. Actually, `A2-winner-v` is able to outperform baselines in 16 out of 18 environments (with 15 unseen *test environments*), while `A2-winner` only outperforms baselines in 12 out of 18 environments. However, please note that it costs another 5k GPU hours (on NVIDIA P100) to

Table 5: Statistical analysis on auxiliary loss functions. The number reported is the difference of the expected RL score when the auxiliary losses *have* one pattern compared to those *do not have*. The corresponding p-value from the t-test is also reported. Positive numbers indicate that this pattern is beneficial. If the performance gain is statistically significant, the number is marked with the asterisk, indicating it is very likely to be helpful. Negative numbers indicate this pattern is detrimental.

| | The score difference between average performances w/ and w/o typical patterns (w/ - w/o) | | | | |
| --- | --- | --- | --- | --- | --- |
| | Forward dynamics | Inverse dynamics | Reward prediction | Action inference | State reconstruction |
| Cheetah-Run (Image) | +1.28 | −3.51 | −31.16** | −75.95** | +42.44** |
| Reacher-Easy (Image) | +28.25* | +8.36 | +37.80** | +3.35 | +70.72** |
| Walker-Walk (Image) | +22.20 | −48.59** | −8.11 | +29.86* | +13.93 |
| Cheetah-Run (Vector) | +94.18** | −23.66** | −33.28** | −109.33** | −50.15** |
| Hopper-Hop (Vector) | +15.50** | −16.47** | −11.30* | −32.10** | −25.67** |
| Quadruped-Run (Vector) | −28.07 | −18.19 | −114.23** | −105.37** | −82.06** |

∗: p-value $< 0.05$. ∗∗: p-value $< 0.01$

| | The score difference between two sets varying the number of elements in source and target | | |
| --- | --- | --- | --- |
| | State, $n_{target} > n_{source}$ | Action, $n_{target} > n_{source}$ | Reward, $n_{target} > n_{source}$ |
| Cheetah-Run (Image) | +80.09** | +13.62 | +3.33 |
| Reacher-Easy (Image) | +1.98 | −12.72 | +65.66** |
| Walker-Walk (Image) | +73.56** | +42.22* | −41.90* |
| Cheetah-Run (Vector) | +188.06** | −102.62** | −93.94** |
| Hopper-Hop (Vector) | +19.80** | −29.70** | −5.03 |
| Quadruped-Run (Vector) | +75.17** | −4.31 | −46.60* |

∗: p-value $< 0.05$. ∗∗: p-value $< 0.01$

search for `A2-winner-v` while there is no additional cost to directly use `A2-winner`. It is a trade-off between lower computational cost and better performance.

## 6 Analysis of Auxiliary Loss Functions

In this section, we analyze all the loss functions we have evaluated during the evolution procedure as a whole dataset in order to gain some insights into the role of auxiliary loss in RL performance. By doing so, we hope to shed light on future auxiliary loss designs. We will also release this "dataset" publicly to facilitate future research.

**Typical patterns.** We say that an auxiliary loss candidate has a certain pattern if the pattern's *source* is a subset of the candidate's *source*, and the pattern's *target* is a subset of the candidate's *target*. For instance, a loss candidate of $\{s_t, a_t\} \rightarrow \{s_{t+1}, s_{t+2}\}$ has the pattern $\{s_t, a_t\} \rightarrow \{s_{t+1}\}$, and does not have the pattern $\{a_t, s_{t+1}\} \rightarrow \{s_t\}$. We then try to analyze whether a certain pattern is helpful to representation learning in RL in expectation.

Specifically, we analyze the following patterns: (i) forward dynamics $\{s_t, a_t\} \rightarrow \{s_{t+1}\}$; (ii) inverse dynamics $\{a_t, s_{t+1}\} \rightarrow \{s_t\}$; (iii) reward prediction $\{s_t, a_t\} \rightarrow \{r_t\}$; (iv) action inference $\{s_t, s_{t+1}\} \rightarrow \{a_t\}$ and (v) state reconstruction in the latent space $\{s_t\} \rightarrow \{s_t\}$. For each of these patterns, we categorize all the loss functions we have evaluated into (i) *with* or (ii) *without* this pattern. We then calculate the average RL performances of these two categories, summarized in Table 5. Some interesting observations are as follows.

(i) Forward dynamics is helpful in most tasks and improves RL performance on Reacher-Easy (image) and Cheetah-Run (vector) significantly (p-value$<$0.05).

(ii) State reconstruction in the latent space improves RL performance in image-based tasks but undermines vector-based tasks. The improvements in image-based tasks could be attributed to the combination of augmentation techniques, which, combined with reconstruction loss, enforces the extraction of meaningful features. In contrast, no augmentation is used in the vector-based setting, and thus the encoder learns no useful representations. This also explains why CURL performs poorly in vector-based experiments.

(iii) In the vector-based setting, some typical human-designed patterns (e.g., reward prediction, inverse dynamics, and action inference) can be very detrimental to RL performance, implying that some renowned techniques in loss designs might not work well under atypical settings.

**Number of Sources and Targets.** We further investigate whether it is more beneficial to use a small number of sources to predict a large number of targets ($n_{target} > n_{source}$, e.g., using $s_t$ to predict $s_{t+1}, s_{t+2}, s_{t+3}$), or the other way around ($n_{target} < n_{source}$, e.g., using $s_t, s_{t+1}, s_{t+2}$ to predict $s_{t+3}$). Statistical results are shown in Table 5, where we find that auxiliary losses with more states

on the *target* side have a significant advantage over losses with more states on the *source* side. This result echoes recent works [42, 39]: predicting more states leads to strong performance gains.

## 7 Related Work

**Reinforcement Learning with Auxiliary Losses.** Usage of auxiliary tasks for learning better state representations and improving the sample efficiency of RL agents, especially on image-based tasks, has been explored in many recent works. A number of manually designed auxiliary objectives are shown to boost RL performance, including observation reconstruction [51], reward prediction [20], dynamics prediction [6] and contrastive learning objectives [23, 39, 42]. It is worth noting that most of these works focus on image-based settings, and only a limited number of works study the vector-based setting [32, 35]. Although people may think that vector-based settings can benefit less from auxiliary tasks due to their lower-dimensional state space, we show in our paper that there is still much potential for improving their performance with better learned representations.

Compared to the previous works, we point out two major advantages of our approach. (i) Instead of handcrafting an auxiliary loss with expert knowledge, A2LS automatically searches for the best auxiliary loss, relieving researchers from such tedious work. (ii) A2LS is a principled approach that can be used in arbitrary RL settings. We discover great auxiliary losses that bring significant performance improvement in image-based and the rarely studied vector-based settings.

**Automated Reinforcement Learning.** RL training is notoriously sensitive to hyper-parameters and environment changes [18]. Recently, many works attempted to take techniques in AutoML to alleviate human intervention, for example, hyper-parameter optimization [7, 36, 49, 53], reward search [9, 45] and network architecture search [38, 10]. In contrast to these methods which optimize a new configuration for each environment, we search for auxiliary loss functions that generalize across different settings such as (i) different robots of control; (ii) different data types of observation; (iii) partially observable settings; (iv) different network architectures; (v) different benchmark domains.

**Automated Loss Design.** In the AutoML community, it has become a trend to design good loss functions that can outperform traditional and handcrafted ones. To be specific, to resolve computer vision tasks, AM-LFS [27] defines the loss function search space as a parameterized probability distribution of the hyper-parameters of softmax loss. A recent work, AutoLoss-Zero [28], proposes to search loss functions with primitive mathematical operators.

For RL, existing works focus on searching for a better RL objective, EPG [19] and MetaGenRL [21] define the search space of loss functions as parameters of a low complexity neural network. Recently, [4] defines the search space of RL loss functions as a directed acyclic graph and discovers two DQN-like regularized RL losses. Note that none of these works investigates auxiliary loss functions, which are crucial to facilitate representation learning in RL and to make RL successful in highly complex environments. To the best of our knowledge, our work is the first attempt to search for auxiliary loss functions that can significantly improve RL performance.

## 8 Conclusion and Future Work

We present A2LS, a principled and universal framework for automated auxiliary loss design for RL. By searching on *training environments* with this framework, we discover a top-performing auxiliary loss function A2-winner that generalizes well to a diverse set of *test environments*. Furthermore, we present an in-depth investigation of the statistical relations between auxiliary loss patterns and RL performance. We hope our studies provide insights that will deepen the understanding of auxiliary losses in RL, and shed light on how to make RL more efficient and practical. Limitations of our current work lie in that searching requires an expensive computational cost. In the future, we plan to incorporate more delicate information such as higher-order information [12] of the inner-loop RL training procedure to derive more efficient auxiliary loss search methods.

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
