# OpenReview forum: "Reinforcement Learning with Automated Auxiliary Loss Search"
_NeurIPS.cc/2022/Conference — NeurIPS 2022 Accept_

### Official Review · Reviewer_bXkd · 2022-07-11

**Rating:** 7
**Confidence:** 5
**Soundness:** 3 good
**Presentation:** 4 excellent
**Contribution:** 3 good

**Summary:**

The paper proposes to use an evolutionary strategy to search the best auxiliary losses for reinforcement learning. The main technical contribution is the design of a search space that covers a set of common auxiliary losses such as the forward dynamics, inverse dynamics and reward prediction. Empirical experiments show that, a loss searched on 3 image-based DM Control tasks also brings better sample efficiency on other DM Control tasks and Atari games.

**Questions:**

Other suggestions:
1. Eqn. 1 seems a bit odd. Maybe it's better to write it as $\max_{\mathcal{L}_{aux}} \mathcal{R}(\min_w \text{losses})$

2. The normalized AUC score is not explained. Normalized w.r.t. what?

**Limitations:**

Please see above. Since searching over auxiliary losses is computationally expensive and unlikely for other researchers to search for each task they are interested in, I think it is important to state clearly when does the searched auxiliary task transfer and when it does not.

**Strengths And Weaknesses:**

### Strengths
* I like the overall direction of the paper. It did a good job exploring whether we can find a good set of auxiliary losses that transfer to different tasks. It also provides insights (through empirical experiments) what are important auxiliary tasks.
* The experiments are extensive well presented to support the main claims of the paper. The generalization of the searched auxiliary loss are demonstrated on different tasks as well as on different RL algorithms (SAC and Efficient Rainbow) and different input observation.

### Weakness
1. A main question that is not fully addressed in the paper is, to what degree is the searched auxiliary loss (i.e. A2-winnder) universal? When does the auxiliary loss transfer and when does it not? To provide two angles to explore in this aspect:
    * The searched loss in Eqn. 3 involves states up to 3 time steps into the future. Where does this horizon of 3 come from? In another environment where the time is discretized differently, either more frequently or less frequently, should this loss be adapted accordingly? One experiment that can be done is, for the currently tested environment, if we add action repeat i.e. repeat each action for two steps, would the optimal auxiliary loss change?
    * The tasks evaluated in this paper are all 2D tasks. How about the more complicated 3D manipulation tasks, such as meta-world [1],
which is also commonly used for evaluating RL algorithms?

2. Some related works are missing. There are a body of works on balancing multiple auxiliary tasks for reinforcement learning [2,3,4]. These works discuss the interplay between the auxiliary tasks and the main task, as well as among the auxiliary tasks. Specifically, Lin et al. [2] also considers a set of auxiliary tasks are also considered, including the forward and inverse dynamics. Lin et al. [2] proposes to tune the weights of the auxiliary tasks based on the main task performance. In a way, it provides a more efficient way of evaluating an auxiliary loss, and also provides an extention of the search space in the submitted paper by including the weights of the auxiliary losses.

I am willing to raise my score if these concerns can be sufficiently addressed.

[1] Yu, Tianhe, et al. "Meta-world: A benchmark and evaluation for multi-task and meta reinforcement learning." Conference on robot learning. PMLR, 2020.

[2] Lin, Xingyu, et al. "Adaptive auxiliary task weighting for reinforcement learning." Advances in neural information processing systems 32 (2019).

[3] Shi, Baifeng, et al. "Auxiliary task reweighting for minimum-data learning." Advances in Neural Information Processing Systems 33 (2020): 7148-7160.

[4] Chen, Zhao, et al. "Gradnorm: Gradient normalization for adaptive loss balancing in deep multitask networks." International conference on machine learning. PMLR, 2018.

---

> ### Author Response · Authors · 2022-08-01
> **Author Reply to Reviewer bXkd (1/2)**
>
> We are pleased to see your valuable discussions and we try to address your concerns in the rebuttal phase as below.
> ***
> **Q1**: “ to what degree is the searched auxiliary loss (i.e. A2-winner) universal? When does the auxiliary loss transfer and when does it not? When does the auxiliary loss transfer and when does it not?”
>
> **A1**: We really appreciate your valuable question about the limit of generalizability of our discovered auxiliary loss A2-winner.
>
> We define a *successful transfer of A2-winner* as *A2-winner brings significant performance improvement to base RL algorithm*.
>
> In the paper, our generalization/transfer experiments in Section 5 have illustrated that A2-winner can be successfully transferred (from three training environments) to various scenarios, including (i) different control tasks; (ii) different robots; (iii) different data types of observation; (iv) partially observable settings; (v) different network architectures; (vi) different discrete control domain (Atari); (vii) different RL algorithms.
>
> From all the experiments we have conducted, we find that our discovered auxiliary loss function A2-winner has great generalizability in these novel settings. This indicates the great generalizability of the searched auxiliary loss.
>
> Furthermore, we also analyzed why the generalization ability of A2-winner is strong in Section 6. As Table 5 suggests, there are some typical patterns of auxiliary losses for RL. Patterns like (i) forward dynamics and  (ii) $n_{target} > n_{source}$ (which A2-winner has) are beneficial in most environments. And we believe that it provides some evidence and reasons why the generalization ability of A2-winner is good in most transferring settings.
> Although A2-winner has already significantly outcompeted baselines on various settings, searching from scratch using our proposed method for a new kind of task can even further improve the performance, yet it can be expensive as you also mentioned. The trade-off between lower computational cost and better performance has been discussed in the paragraph "to search or not" paragraph (Line 261 to 272) in Section 5.
> ***
>
> Next, we make further experimental exploration and discussion about two settings you mentioned in the review comments.
>
> **Q1.1**: “The searched loss in Eqn. 3 involves states up to 3 time steps into the future. … In another environment where the time is discretized differently, either more frequently or less frequently, should this loss be adapted accordingly?”
>
> **A1.1**: We believe this is really a good point that is worth discussing. Actually, in our experiments on DMControl, there are already many environments with a different number of action repeats, as reported in Table 6 (Walker: 2; Cheetah: 4; Cartpole: 8), and A2-winner consistently performs well across all these environments.
> As reported in Section F.2, there are many top-performing auxiliary losses with different numbers of time steps into the future. We finalize A2-winner because (i) A2-winner is sufficiently good enough and (ii) the budget of computing resources.
> In fact, we do not claim that A2-winner is the *optimal* auxiliary loss for RL. As shown in Figure 4, we believe there is a set of top-performing auxiliary losses that transfer well to different tasks, while A2-winner is one of them.
>
> To further address your concerns on the optimality of *horizon* (i.e., time steps in the future), we analyze the correlation between *horizon* and RL performance using all data during evolution. As shown in the [histogram](https://anonymous.4open.science/r/A2LS-NeurIPS22-Rebuttal-C575/analysis_horizon.pdf), we count the average AULC scores for three categories: (i) horizon <= 2; (ii) 3 <= horizon <=5; (iii) 5 <= horizon. We conclude that no specific range of horizon is statistically superior. Discussing the optimal horizon of auxiliary losses would be an interesting topic for future works.
>
> To further address your concerns on the limit of the generalization ability of A2-winner to different time discretization, we conduct experiments to test A2-winner with a different number of action repeating (2/4/8) times in one timestep on Cheetah-run, and we believe the repeating action setting has imitated the settings with different timestep discretization to some extent. As shown in the [figure](https://anonymous.4open.science/r/A2LS-NeurIPS22-Rebuttal-C575/ablation_cheetah-run_action_repeat-crop.pdf), A2-winner again significantly outperforms CURL and SAC in all settings of action repeats, which reflects the good generalization ability of our discovered auxiliary loss in this various timestep discretization settings.

---

> > ### Author Response · Authors · 2022-08-01
> > **Author Reply to Reviewer bXkd (2/2)**
> >
> > **Q1.2**: “The tasks evaluated in this paper are all 2D tasks. How about the more complicated 3D manipulation tasks, such as meta-world…”
> >
> > **A1.2**: To test the generalization ability of A2-winner to 3D tasks, we implement A2-winner for meta-world benchmarks. We select 4 challenging 3D tasks in meta-world. As shown in [the meta-world result](https://anonymous.4open.science/r/A2LS-NeurIPS22-Rebuttal-C575/meta_world.pdf), A2-winner has also significantly outperformed the other baselines (SAC and CURL) across 4 environments.
> >
> > Though we have evaluated the searched auxiliary loss function A2-winner in various different scenarios and settings and illustrated that applying our searching method in novel environments can further improve the performance in spite of additional computational costs, we believe there are some limitations in the regime of the RL research. For example, we conducted the experiments in the online RL scenario yet we have not explored our method in offline RL settings. Offline RL shares similar settings such as MDP assumption and maintaining the collected transition data, however, it also illustrates different properties compared to online RL. We can either transfer our searching algorithm to that scenario or just implement the searched results onto that in our future work.
> >
> > In general, we believe it's important for future studies to further analyze how and why our searched result A2-winner auxiliary loss performs well (or not well) in more settings, as you suggested.
> >
> > ***
> > **Q2**: “Some related works are missing….”
> >
> > **A2**: We are sorry for the missing reference, and thanks for your sincere advice. In our revision, we have added another literature review section (see Appendix Section G) on works balancing multiple auxiliary tasks for reinforcement learning, which can be further refined and merged into the main text in the future version.
> >
> >
> > We really appreciate your suggestion on balancing auxiliary losses and RL. Actually, we have considered including the loss weights $\lambda$ of the auxiliary losses into our search space at the beginning. But during our experiments, we find that RL performance is actually robust to different values of the loss weight $\lambda$ (shown in Eqn. 1), see [cheetah experiment](https://anonymous.4open.science/r/A2LS-NeurIPS22-Rebuttal-C575/AuxiCoef_cheetah-run.pdf). We will add more discussion on the relative weights of the auxiliary losses in the future version.
> >
> >
> >
> > ***
> > **Q3**: “​​Eqn. 1 seems a bit odd. Maybe it's better to write it as”
> >
> > **A3**: Thanks for your advice. We have revised Eqn. 1 to $\max_{\mathcal{L_{aux}}} \mathcal{R}(\min_{w} losses)$ in our revised paper.
> >
> > ***
> > **Q4**: “The normalized AULC score is not explained. Normalized w.r.t. what?”
> >
> > **A4**: We are sorry for the confusion. The normalized AULC score is normalized with respect to the lowest AULC score among candidates during cross validation. For example, suppose we have 5 candidates with AULC scores of $13, 14, 18, 12 (lowest), 15$, and then the normalized AULC of candidate-4 will be set as $1$. We have updated Figure 5 for clarity in the revised paper. Note that the normalized AULC score is only used in cross validation experiments to better illustrate the performance comparison among different top-performing candidates in different training environments.

---

> > > ### Comment · Reviewer_bXkd · 2022-08-07
> > > **Thank you**
> > >
> > > I have read comments and responses from all the reviews. I thank the authors for their responses and additional experiments. I appreciate the authors for providing more experiments in such a short time. They have addressed most of my concerns about the related works and the transferability of the proposed method.  I have raised my score accordingly.
> > >
> > > On the other hand, my concern about transferring the A2-winner to another time discretization was more on the theoretical side. If the 3-step prediction into the future is more related to the RL algorithm itself, including the hyperparameters, then I would assume A2-winner can transfer to a different time discretization without changes. On the other hand, if the 3-step prediction is related to the control frequency of the environment and the tasks, then it should be adjusted accordingly under a different time discretization. I thank the authors for providing experiments on different action repeats. But I think the more interesting comparison is to test A2-winnder with differnet n-step prediction and see if the best n varis under different action repeats.

---

> > > > ### Author Response · Authors · 2022-08-08
> > > > **Thank you for your response**
> > > >
> > > > We agree that testing A2-winner with different n-step predictions is an interesting direction to investigate whether the 3-step prediction is more related to the control frequency or the RL algorithm itself. Due to the time limit of rebuttal, we will add more discussion on this in future versions.
> > > >
> > > > Thank you again for your reply and helpful suggestions.

---

### Official Review · Reviewer_Jaq4 · 2022-07-11

**Rating:** 8
**Confidence:** 3
**Soundness:** 3 good
**Presentation:** 4 excellent
**Contribution:** 4 excellent

**Summary:**

The paper proposes a method to do automated search for reinforcement learning auxiliary losses. This is done by formulating a configuration for possible losses, which covers many of the common losses used in the field (e.g. contrastive, self-predictive), and searching using evolutionary search. This leads to a bi-level optimization: outer loop -- proposing, selecting and evolving candidate loss functions -- and inner loop, where the candidate loss function is evaluated by training an RL agent together with the candidate loss, and using the overall performance as a metric for the outer loop.

The method is applied to continuous control DeepMind Control Suite, where an auxiliary loss (A2-winner) is found that consistently outperforms the baselines. Moreover, this loss has similarly strong results on different test environments, including e,g, some with different observation types or discrete control domain (Atari).

**Questions:**

In addition to the points above (including information about the inner loop algorithm in the main paper and adding a short discussion on how the baselines were chosen), the following aspects might be useful to consider:
- How does the method compare with other AutoML methods applied to the task?
- Do the findings apply when tested in the setting of task generalization?
- What values of lambda (eq 1) were tried and how do the results compare?
- Section 3.2.1 provides a very specific method to prune the search space, seemingly inspired by empirical evidence, rather than a general assumption. More discussion on results/generality could make its scope clearer.
- Does the search space include reconstruction losses at the moment?

**Limitations:**

The authors have adequately noted and addressed the limitations of the proposed method.

**Strengths And Weaknesses:**

The paper has many strong aspects: clarity, thorough evaluation of the proposed candidate loss on many different test environments and originality in applying the AutoML framework to the space of auxiliary losses for RL and in, among others, proposing a loss configuration that covers most of the hand-engineered loss functions, as well as being interpretable (the proposed A2-winner seeming like a combination of reward prediction and forward dynamics).

The method is expensive, requiring tens of thousands of GPU hours, and there is room for improvements in efficiency, which is also remarked by the authors. However, this does not represent a weakness with respect to the work's significance, as important insights can already be drawn from the presented empirical evidence and, further, the results of A2-winner loss seems to generalize to environments on which the search was not performed.

However, presenting more information about the inner loop and its comparison with the baselines (hyper-parameters, choosing criteria) could be highly beneficial to leveraging the work in the future and answering questions such as: do these findings apply to value-based, as well as policy-based methods?

Overall, as most aspects are highly positive, the paper meets the acceptance criteria.

---

> ### Author Response · Authors · 2022-08-01
> **Author Reply to Reviewer Jaq4 (1/2)**
>
> We sincerely thank you for your comprehensive comments and valuable suggestions on our paper. We have revised our paper based on your valuable reviews in the revision. And we are pleased to respond to your inspiring questions as below.
> ***
> **Q1**: “presenting more information about the inner loop and its comparison with the baselines (hyper-parameters, choosing criteria) could be highly beneficial to leveraging the work in the future”
>
> **A1**: We agree with your valuable suggestions. In the revised paper, we have supplemented information about (i) inner loop algorithm in Appendix Section B (ii) hyper-parameters of baselines in Appendix Section C.4; (iii) choosing criteria and implementation of baselines in Appendix Section C.5.  Due to the 9-page limitation of NeurIPS submission, we can only add references in the main text for now. After all other revisions are complete, we will move the information about the inner loop and baselines to the main text to improve the readability of the future version.
> ***
> **Q2**: “...could be highly beneficial to leverage the work in the future and answer questions such as: do these findings apply to value-based, as well as policy-based methods?”
>
> **A2**: In the revised paper, we have tried to answer the above question (in line 231 in Section 5) that our findings generalize well to both value-based (DQN Rainbow) and policy-based RL (SAC) methods.
> ***
> **Q3**: “How does the method compare with other AutoML methods applied to the task?”
>
> **A3**: Typical AutoML methods include: (i) gradient-based optimization; (ii) reinforcement learning; (iii) surrogate model-based optimization; (iv) grid and random search (v) evolutionary search. We think it is inspiring to use other AutoML methods to explore the search space we proposed.
> But to the best of our knowledge, our work is the first attempt to search for auxiliary loss functions that can significantly improve RL performance using AutoML techniques. The setup is new, and thus there are no strictly comparable AutoML methods. Nevertheless, we compare random sampling with our method (details are given in the reply of *A2* and *A3* to reviewer qgb7), showing the superiority of our work.
> ***
> **Q4**: “Do the findings apply when tested in the setting of task generalization?”
>
> **A4**: Thanks for your inspiring suggestion! According to the great generalization ability of the A2-winner verified in Section 5, we believe our findings (in Section 6) of beneficial loss patterns might still apply to other tasks.
> It would be interesting to empirically verify our findings about beneficial/detrimental patterns of auxiliary losses in the setting of task generalization, but this requires re-evaluating a population of auxiliary losses on many different tasks, in order to draw the conclusion whether auxiliary losses with a certain pattern are better than those without. We estimate the computational cost to be similar to a loss search from scratch for dozens of new tasks.
> It could be a future research direction to figure out more efficient and systematic approaches to summarize helpful loss patterns.

---

> > ### Author Response · Authors · 2022-08-01
> > **Author Reply to Reviewer Jaq4 (2/2)**
> >
> > **Q5**: “What values of lambda (eq 1) were tried and how do the results compare?”
> >
> > **A5**: **The $\lambda$ in (eq 1) is fixed as 1 across all experiments (following exactly the same setting as CURL)**. To clarify this we have added $\lambda$ to Table 6 and Table 7 in our revision. During our experiments, we find that RL performance is actually robust to different $\lambda$ (eq 1), see [cheetah experiment](https://anonymous.4open.science/r/A2LS-NeurIPS22-Rebuttal-C575/AuxiCoef_cheetah-run.pdf).
> > ***
> > **Q6**: “Section 3.2.1 provides a very specific method to prune the search space, seemingly inspired by empirical evidence, rather than a general assumption. More discussion on results/generality could make its scope clearer”
> >
> > **A6**: Thanks for your valuable advice! We have included some discussion of search space pruning in Section D.1. In our paper, since the search space pruning can be combined with *input elements* and *operator*, we regard it intuitively to decouple the effect of these two types of loss elements.
> > In our early experiments, we verified the assumption that the decoupling of *input elements* and *operator* is reasonable. We conducted experiments in Cheetah-Run with three different auxiliary loss types and five different operators. We conclude from the [figure](https://anonymous.4open.science/r/A2LS-NeurIPS22-Rebuttal-C575/pruning_operators.pdf) that: (i) as for the same auxiliary loss type (same x-axis), the RL performance of different operators do not change dramatically  (around ±10%). (ii) the relative ranking of different operators across different auxiliary losses is consistent.
> > In our revision, we further include the above experiment to support search space pruning. Besides, the ablation results reported in Figure 7 (in Section D.1) also validate that pruning improves the evolution process, making it easier to find good candidates.
> >
> > ***
> > **Q7**: “Does the search space include reconstruction losses at the moment?”
> >
> > **A7**: Yes, the search space includes reconstruction loss (e.g., source=$s_t$ and target = $s_t$) in the latent space, and we summarize it in Table 1.

---

### Official Review · Reviewer_qgb7 · 2022-07-15

**Rating:** 5
**Confidence:** 4
**Soundness:** 2 fair
**Presentation:** 3 good
**Contribution:** 2 fair

**Summary:**

This paper proposes an evolutionary strategy (ES)-based auxiliary loss search for training neural networks for RL. The main motivation is that while it is known that auxiliary losses can improve training by aiding useful representation learning, it is not clear which formulation of the auxiliary loss is most beneficial since the number of potential possibilities is very large. Therefore, it can be valuable to automate the search through this space in order to identify the most useful loss functions. The paper proposed to use binary-encoded selection variables for the inputs and targets (called *input elements*) of the auxiliary loss, with associated mutation operations for the ES. For the functional form of the loss itself (called *operator*), the paper does not use a search but instead uses simple random sampling to identify that the MSE loss works best when the inputs and targets selection are such that the auxiliary loss becomes forward dynamics.

Fixing the loss to MSE and using the ES to search for the input elements, the authors identify an auxiliary loss using 3 image-based control environments for training, that leads to improved performance on multiple image-based test environments. It is also able to improve performance on 12 out of 18 tested non-visual RL tasks showing that it can generalize to a variety of tasks. The authors also present aggregated analyses of the losses evaluated during the search procedure to identify common useful patterns among loss functions that improve or reduce performance.

**Questions:**

Did I understand correctly above that A2-winner was obtained with 5 generations of the ES? If so, can the authors offer convincing arguments that the proposed ES is necessary, and address related comments above?

Is appears that at 500K, the performance with A2-winner might not be too different (statistically) from CURL. Is that correct?

**Limitations:**

Yes

**Strengths And Weaknesses:**

The main strength of this paper is that it is the first study to show the feasibility of automating the search of auxiliary loss functions for RL with results indicating that a useful and novel loss function was indeed discovered through the proposed method, with a relatively simple evolutionary strategy design.

I also appreciate that the authors attempted extensive generalization experiments on several unseen environments and compared to strong baselines.

I have some concerns that reduce my confidence in accepting this paper:

- The biggest concern is that it appears that the identified A2-winner loss is obtained with just 5 generations of the proposed ES (the term *stages* appears to be used in the paper, which I found a little confusing without explanation). This makes me wonder if the ES is actually necessary, and whether a random sampling of 4 or 5 times the population size would already produce equally good results, while being more parallelizable. The authors comment in Sec. 4.1 that "Judging from the trend, we believe the performances could improve even more if we had further increased the budget", but it unclear to me from Fig. 4 that there really is a trend of increasing performance, especially in terms of the best individual in the population improving across the three environments.

- In Table 2, many results for A2-winner at 500K steps are in bold, but their difference from CURL does not appear to be statistically significant.

- The main paper omits any discussion of how the target encoder parameters are obtained. This is left completely to the supplementary material, and confused me for the first couple of readings.

- The paper is notoriously missing references from the ES literature, focusing only on references to some recent papers that apply ES to RL.

---

> ### Author Response · Authors · 2022-08-01
> **Author Reply to Reviewer qgb7 (1/3)**
>
> We sincerely thank you for your comprehensive comments on our paper. And we will try to address all the concerns as below.
>
> ***
> **Q1**: “the term *stage* … I found a little confusing without explanation”
>
> **A1**: We are sorry for the confusion. In our revision, we have added some explanations of *stage*  in the revised paper (see line 143 in Section 3.2). Briefly speaking, *stage* refers to the evolved generation that the objective population is located. Here “population” means the maintained auxiliary loss functions. As has been discussed in Section 3.2.2, at each stage, the maintained auxiliary loss functions will take mutation and move to the next stage.
>
> ***
> **Q2**: “A2-winner loss is obtained with just 5 generations of the proposed Evolutionary Search (ES) … This makes me wonder if the ES is actually necessary, and whether a random sampling of 4 or 5 times the population size would already produce equally good results … but it is unclear to me from Fig. 4 that there really is a trend of increasing performance.”
>
> **A2**: We are sorry for the confusion. In the revision, we supplement additional results in Section D.11 to ease this concern. Specifically, to illustrate the trend of increasing performance during evolution, we provide the *average AULC score of populations of each stage* as follows.
>
> |  AULC | stage-1  | stage-2  | stage-3  | stage-4  | stage-5  | stage-6  | stage-7 | SAC (baseline) |
> |  :---                |    :----:    |    :----:    |    :----:    |    :----:    |    :----:    |    :----:     |   :----:     |  :----: |
> | Cheetah-Run | 191.75  | 252.51   | 258.09   | 284.53   | 349.52   | 351.51   | 352.57   | 285.82 |
> | Reacher-Easy | 674.87  | 782.75   | 812.61   | 823.04   | 810.15   | 811.88   | 827.19  | 637.60 |
> | Walker-Walk | 599.38   |  633.75   |  716.18 |  702.49  |  N/A       |  N/A       |  N/A       | 675.84 |
>
>
> As for comparing ES with random sampling, we can take **stage-1 of each evolution procedure as random sampling**. As shown in the above tabular, the average performance of the stage-1 population (i.e., random sampling) is even worse than SAC in Cheetah-Run and Walker-Walk. Nevertheless, as the evolution continues, the performance of the evolved population in the following stages **improves significantly**, surpassing the score of SAC.
>
> To address your concern on *whether the best individuals in the population are improving or not*, we provide the **average AULC score of the top 5 candidates of the population at each stage** as follows.
>
> | AULC | stage-1  | stage-2  | stage-3  | stage-4  | stage-5  | stage-6  | stage-7 | SAC (baseline) |
> |  :---                |    :----:    |    :----:    |    :----:    |    :----:    |    :----:    |    :----:     |   :----:     |  :----: |
> | Cheetah-Run | 398.18  | 424.27   | 428.08   | 485.54   | 487.94   | 482.65   | 498.46   | 285.82 |
> | Reacher-Easy | 931.27  | 950.61   | 943.83   | 938.91   | 954.77   | 955.02   | 969.43  | 637.60 |
> | Walker-Walk | 834.09   |  883.77   |  896.52 |  880.73  |  N/A       |  N/A       |  N/A       | 675.84 |
>
> As shown above, there is an **obvious trend** that the performance of the best individuals in the population at each stage continues to improve and also outperforms the baseline by a large margin during the evolution across all three training environments.

---

> > ### Author Response · Authors · 2022-08-01
> > **Author Reply to Reviewer qgb7 (2/3)**
> >
> > **Q3**: “... can the authors offer convincing arguments that the proposed ES is necessary?”
> >
> > **A3**: A more intuitive example is shown in this [figure](https://anonymous.4open.science/r/A2LS-NeurIPS22-Rebuttal-C575/jump_of_top5_curve_evolution.pdf), which illustrates the curve of top-5 AULC scores during evolution, where (i) the x-axis represents the number of auxiliary losses evaluated during the evolution *(1-100 corresponds to stage-1, 101-200 corresponds to stage-2, etc.)* and  (ii) the y-axis represents the mean value of the top-5 AULC scores *so far* in the population. At each red point which represents the end of the previous stage and the start of the next successive stage, two processes of evolution take place: (i) evaluation and selection (top 25%) on the population and (ii) mutation (i.e., producing new auxiliary losses) *based on the selected population*.
> >
> > Note that the difference between pure random sampling and evolutionary search is that random sampling does not have the procedure of *selecting top candidates and mutating for new generations*. Therefore, we can understand the first stage of evolutionary search as random sampling.
> >
> > **We can see from the [curve](https://anonymous.4open.science/r/A2LS-NeurIPS22-Rebuttal-C575/jump_of_top5_curve_evolution.pdf) that, before the point of 100 (i.e., random sampling), the learning curve is converging. After 100, there is a jump in the evolution curve, which exactly corresponds to the effect of the evolution procedure between stage-1 and stage-2, proving that evolution generates stronger top-performing candidates.**
> > Therefore, obtaining top-performing candidates from the evolution is more efficient than random sampling since the evolution search gives stronger candidates. We believe that (i) the trend of improving scores reported in the above tables and (ii) the [top AULC value jump](https://anonymous.4open.science/r/A2LS-NeurIPS22-Rebuttal-C575/jump_of_top5_curve_evolution.pdf)  at each time when the next evolution stage begins have demonstrated the necessity of evolutionary search.
> >
> > Ideally, fairly comparing evolutionary search against random search requires running another random search experiment with the same budget as the evolutionary search. But such an experiment requires another expensive GPU cost that is hard to finish during the rebuttal period. Nevertheless, the efficiency advantage of evolutionary search over random sampling has also been verified in many AutoML literatures [1,2,3].
> >
> > [1] Co-Reyes, J. D., Miao, Y., Peng, D., Real, E., Levine, S., Le, Q. V., ... & Faust, A. (2021). Evolving reinforcement learning algorithms. arXiv preprint arXiv:2101.03958.
> >
> > [2] Houthooft, R., Chen, Y., Isola, P., Stadie, B., Wolski, F., Jonathan Ho, O., & Abbeel, P. (2018). Evolved policy gradients. Advances in Neural Information Processing Systems, 31.
> >
> > [3] Whitley, D., Rana, S., Dzubera, J., & Mathias, K. E. (1996). Evaluating evolutionary algorithms. Artificial intelligence, 85(1-2), 245-276.

---

> > > ### Author Response · Authors · 2022-08-01
> > > **Author Reply to Reviewer qgb7 (3/3)**
> > >
> > > **Q4**: “In Table 2, many results for A2-winner at 500K steps are in bold, but their difference from CURL does not appear to be statistically significant.”
> > >
> > > **A4**: Thank you for pointing this out. As a matter of fact, we have tested A2-winner on a total of 12 image-based DMC tasks (as illustrated by Figure 8 in Appendix D.2) but only reported performance on 6 tasks in Table 2 following the practice of prior work [4]. It turns out that the 6 tasks are the relatively easier ones, and both A2-winner and CURL converge to near-optimal (near 1000 score) at 500k environment steps. But still, A2-winner outperforms baselines significantly at 100k steps, indicating it has a better sample efficiency. While
> > > in [Figure 8](https://anonymous.4open.science/r/A2LS-NeurIPS22-Rebuttal-C575/Figure8_12DMC_env.png) in Appendix D.2, compared to CURL, our method has a much stronger performance in some more challenging tasks (a total of 12 tasks), in which the performance gain is actually significant at 500K and beyond. For example, at 500K, in the cartpole-swingup-sparse task, we reach more than 700 scores while CURL is lower than 400. In hopper-stand, we reach near 800 scores at 500K while CURL is only near 400.
> > > In summary, our method not only has significantly better sample efficiency during the early training, but also has the overall best performance at 500K+ steps on a number of challenging tasks.
> > >
> > > [4] Laskin, M., Srinivas, A., & Abbeel, P. (2020, November). Curl: Contrastive unsupervised representations for reinforcement learning. In International Conference on Machine Learning (pp. 5639-5650). PMLR.
> > >
> > >
> > > ***
> > > **Q5**: “The main paper omits any discussion of how the target encoder parameters are obtained. This is left completely to the supplementary material, and confused me for the first couple of readings.”
> > >
> > > **A5**: Thanks for pointing this out! We have added more descriptions and references (in Section 3.1) to clarify the mechanism of updating the target encoder.
> > >
> > > ***
> > > **Q6**: “The paper is notoriously missing references from the ES literature, focusing only on references to some recent papers that apply ES to RL.”
> > >
> > > **A6**: We are sorry for the missing references, and thanks for your sincere advice. In our revision, we have added another literature review section (see Appendix Section G) on ES literature, which can be further refined and merged into the main text in the future version.
> > > ***
> > > We have revised our paper based on your valuable reviews in the revision.
> > > Please let us know if you have any further comments. We will try our best to address them and improve our paper.

---

> > > > ### Author Response · Authors · 2022-08-08
> > > > **Thanks for your comment. We are willing to address further concerns.**
> > > >
> > > > Dear Reviewer qgb7,
> > > >
> > > > We appreciate your valuable and constructive comments, which provide much helpful guidance to improve the quality of our paper. We hope our previous reply has resolved all your concerns. If you have any other questions, we are also pleased to respond. We sincerely look forward to your response.
> > > >
> > > > Best wishes!
> > > >
> > > > The authors.

---

> > > > > ### Comment · Reviewer_qgb7 · 2022-08-09
> > > > > **Re: Response**
> > > > >
> > > > > Thank you for the response and providing the additional details.
> > > > >
> > > > > Unfortunately the details don't convince me. While I'm very familiar with the ability of evolutionary methods to solve complex optimization problems, it is unclear from the data here that the ES is doing much here. May be it is, it's just not clear. To clarify further: to me the question isn't whether it is _possible_ to search for better auxiliary loss functions, but whether the specific method proposed here is clearly effective. ES, and EAs in general do solve complex search problems, but they are typically not used when it takes only 5 generations or so to find good enough solutions. In such situations, simply increasing the number of random search samples can often give the (best) result of equal quality, while the search is more parallelizable. As you correctly point out, a critical question is: for the same number of total function evaluations, how do the results --- not of the population avg, but at the best solution level --- compare between random sampling and the ES?
> > > > >
> > > > > In black-box optimization literature, not only is this comparison necessary, but one must usually also do repeated comparisons (due to randomness) to understand whether a search technique adds value. I'm sorry about this, and I understand that this requires more compute and time, but that is perhaps the nature of AutoML.

---

> > > > > > ### Author Response · Authors · 2022-08-09
> > > > > > **Comparing Evolutionary Search and Random Sampling**
> > > > > >
> > > > > > **Q**: a critical question is: for the same number of total function evaluations, how do the results --- not of the population avg, but at the best solution level --- compare between random sampling and the ES?
> > > > > >
> > > > > > **A**: We would like to highlight the [curve of search progress](https://user-images.githubusercontent.com/110891305/183591489-7f6cd87d-8083-4e79-9c12-129526612c81.png), with some additional clarifications.  The x-axis is the number of loss candidates trained, while the y-axis represents the average of the best 5 AULC scores ever searched *so far*.
> > > > > >
> > > > > > Due to limitation of computational resources and rebuttal due, we use the following approach to compare evolution against random sampling *under the same budget*. We reuse the results from the first stage of evolution (essentially a random sampling strategy), compute several statistics (i.e., mean and std), and use a gaussian distribution to simulate a random sampling strategy with the same budget as ES.
> > > > > > As shown in the [figure](https://user-images.githubusercontent.com/110891305/183591489-7f6cd87d-8083-4e79-9c12-129526612c81.png), **the line of evolutionary search (blue) is consistently higher than simulated random sampling (orange)**.
> > > > > > More importantly, **after 100/200/300, there are jumps in the evolution curve**, which exactly corresponds to the effect of the evolution procedure between each two successive stages, proving that **evolution effectively generates stronger top-performing candidates.**
> > > > > >
> > > > > > In conclusion, we believe that, the ES strategy is more efficient in finding the top-performing auxiliary losses compared to random sampling strategy, *under the same and limited computational budget*.

---

> > > > > > > ### Comment · Reviewer_qgb7 · 2022-08-10
> > > > > > > **Response**
> > > > > > >
> > > > > > > Thank you for providing the additional plot with the simulated random sampling, but I'm not convinced that this gives a statistically accurate picture of the difference in performance between the proposed method and random sampling. I agree that there are _indications_ it can work better, but given the large amounts of stochasticity and the few number of generations required to reach the results, the indications are not proof and can't be used to draw formal conclusions in my view. Again, my apologies, and I appreciate the hard work to produce these results.

---

> > > > > > > > ### Author Response · Authors · 2022-08-10
> > > > > > > > **Thank you**
> > > > > > > >
> > > > > > > > After pondering on your comments, we agree with you that parallelized random sampling has the potential to achieve even stronger performance. However, we would like to point out that:
> > > > > > > >
> > > > > > > > Our paper has already made a large number of important contributions: (i) we are the first to search for auxiliary loss functions that can significantly improve RL performance; (ii) we develop a complete pipeline for conducting these experiments; (iii) we show that it works well (iv) and we achieve significantly stronger performance while revealing a number of novel insights.
> > > > > > > >
> > > > > > > > We do not claim to have fully solved all research problems in this area and our paper certainly will not be the last paper in this direction. Investigating alternative search methods such as parallelized random sampling is an exciting future research direction. However, this requires a lot of new experiments and GPU hours to prove, and seems to be out of the scope of this paper which already has a significant amount of content.
> > > > > > > >
> > > > > > > > For the problem you mentioned, we will emphasize in our paper that although our empirical results indicate that ES works well in our setting, alternative search strategies should be further investigated to decide which one is the best approach.
> > > > > > > >
> > > > > > > > We have been trying our best in the rebuttal period, and we hope we have eased most of your concerns on (i) increasing trend during evolution; (ii) necessity of evolutionary search; (iii) scores at 500K; (iv) target encoders; (v) missing literature. Thank you again for your helpful review and response.

---

### Author Response · Authors · 2022-08-07
**We sincerely look forward to your reply.**

Dear reviewers,

We first thank you again for your valuable comments and suggestions. In the previous replies, we think we have addressed your questions point by point and added the corresponding experiments.

We sincerely look forward to your reply to our response. And we are open to any discussion to improve our paper.


Best wishes!

The authors.

---

### Meta-Review · Area_Chair_PYk5 · 2022-08-31

**Recommendation:** Accept
**Confidence:** Certain

**Metareview:**

The paper introduces an interesting evolutionary scheme for black-box hyper-parameter optimisation for representation learning, looks like a useful and general tool for optimising RL-agent in case the compute is not a constraint.

**Award:**

No

---

### Decision · Program_Chairs · 2022-09-14

Accept